# Patellar Tendon Structural Adaptations Occur during Pre-Season and First Competitive Cycle in Male Professional Handball Players

**DOI:** 10.3390/ijerph182212156

**Published:** 2021-11-19

**Authors:** Silvia Ortega-Cebrián, Ramon Navarro, Sergi Seda, Sebastià Salas, Myriam Guerra-Balic

**Affiliations:** 1Physiotherapy Department, Facultat Fisioteràpia, Universitat Internacional de Catalunya (UIC), Carrer Josep Trueta, Sant Cugat de Vallès, 08017 Barcelona, Spain; 2Futbol Club Barcelona, Medical Department, Ciutat Esportiva Joan Gamper, Avinguda, Once Setembre, Sant Joan Despí, 08970 Barcelona, Spain; ramonrctb@gmail.com (R.N.); sergi.seda@fcbarcelona.cat (S.S.); sebastia.salas@fcbarcelona.cat (S.S.); 3Faculty of Psychology, Education and Sports Sciences, University Ramon Llull, Spain FPCEE-Blanquerna, 08022 Barcelona, Spain; miriamelisagb@blanquerna.url.edu

**Keywords:** tendinopathy, differential diagnosis, physical fitness, tenocyte, sports

## Abstract

Background: While there is evidence that tendon adapts to training load, structural alterations in the patellar tendon in response to training loads are still unclear. The aim of this study is to identify changes in patellar tendon structure throughout pre-season and after finalizing the first competitive cycle. Methods: Nineteen professional handball players participated in the aforesaid cross-sectional study, in which patellar tendon scan and counter movement jump (CMJ) performance were conducted. Measurements were taken on the first and last day of pre-season training, and at the end of the first competitive cycle. Results: The results revealed that variation on the tendon structure occurred, mainly at the end of pre-season training; for injured tendons this occurred at the proximal (Right *p* = 0.02), distal (Right *p* = 0.01), and (Left *p* = 0.02) tendon, while changes in healthy tendons occurred at the mid (Left *p* = 0.01) and distal tendon (Right *p* = 0.01). At the end of the first competitive cycle, changes were observed in the distal injured tendon (*p* = 0.02). Conclusion: Patellar tendon shows greater structural change after completing pre-season training than at the end of the first competitive cycle, from which it may be inferred that gradual loading during pre-season training allows the tendon to adapt and potentially decrease the onset of patellar tendinopathy.

## 1. Introduction

Patellar tendinopathy remains one of the most frequent pathologies in professional athletes: epidemiology studies report that 44.6% of volleyball players and 31.9% of basketball players suffer from patellar tendinopathy [1]. As for handball, the incidence of patellar tendinopathy in professional players is higher than in youth handball players, though lower than in basketball, volleyball, and roller hockey players [1,2]. Although the prevalence of patellar tendinopathy has been recognized, these figures may be subject to question as incidence is only reported when time-loss from sport practice is necessary, whereas patellar tendon pain in active players is not reported [1,3].

Traditionally, jumping mechanism has been considered the main risk factor for patellar tendinopathy, defining this injury as “Jumper’s knee” [4]. Notwithstanding, recent studies have demonstrated patellar tendinopathy to be a multifactorial, multi-etiological pathology that is not solely susceptible to being aggravated by jumping mechanisms [4]. Clinical features of patellar tendinopathy include localized pain in the tendon along with decreased athletic performance [5,6]. This concomitance can be explained by patellar tendon sensitivity to load changes, whether short-term or long-term load [7], which in turn usually present structural alterations in the tendon. Although only symptomatic cases present with pain and dysfunction necessitating cessation of sport activity [8], histological tendon studies demonstrate that either symptomatic or asymptomatic tendons can show changes in matrix structure, some of which conclude in a complex injury process that often requires medical intervention [8,9].

The tendon’s ability to tolerate loads is individual in nature, although there are common factors that increase risk of tendinopathy [10]. Tendon mechanical properties are known to increase by receiving the optimal load and allowing tissue recovery [11]. When load is excessive, the tissue recovery process does not take effect, anchors are lost between tenocytes, and disruption of tenocyte communication occurs [7,8]. The physiopathological processes of tendinopathy have been widely described; an onset of tendon structure changes in the form of misalignment of collagen fibers and loss of continuity produces tendon degeneration, which can be symptomatic or asymptomatic [6]. 

Diagnosis of patellar tendinopathy is usually confirmed by ultrasound and Magnetic Resonance Imaging (MRI); however, the accuracy of imaging in patellar tendinopathies has been recently questioned [12]. Current diagnostic imaging technology allows for quantification of the amount of altered tendon [13] with Ultrasound Tissue Characterization (UTC). UTC classifies tendon structure by four echo types which refer to tendon integrity and fibrillary disorganization: echo type I (green) describes intact and aligned tendon bundles; echo type II (blue) represents tendon regions that demonstrate mild separation and/or more undulating fascicles; echo type III (red) describes areas with signs of fibrillar disorganization; and echo type IV (black) describes tendon degeneration with severely disorganized fibrillar structure in regions with free fluid or amorphous material [14]. This classification of echo types allows monitoring of tendon integrity during treatment and collagen adaptation to load during a certain period of time, as well as identifying potential pathology in asymptomatic subjects [14]. 

In handball sports, physical abilities such as strength, power, endurance, velocity, agility, and coordination are required at high intensity [2,3]. Excessive storage and release energy mechanisms require patellar tendon to receive shear, tensile, and compressive forces, which are known to increase the risk of patellar tendinopathy [2,15]. When the tendon is subject to great demands, it is believed that possible alterations in tendon structure could impact athletic performance as well. If the tendon structure tends to lose mechanical transducers in the tendon matrix, then decreased accumulation of energy storage and alteration of the stretch and shortening cycle will occur [16]. Hence, it may be assumed that jumping performance could be affected by changes in tendon structure, given that this skill is tested throughout the season to measure lower limb strength and power [17].

In the case of handball players, when and how the patellar tendon matrix adapts to training loads during different periods of the season is unclear. Additionally, there is uncertainty about how potential matrix adaptations are reflected in jumping performance. Among all knee injuries in professional handball players, patellar tendinopathy has a poor prognosis, occurring generally during pre-season and after the first competitive cycle and being associated with increased training and competition load [1,2,3,15]. Quantifying how and when the patellar tendon adapts to training loads during this period of time could provide better understanding of the onset of patellar tendinopathy.

This study aims to compare tendon structure in pre-season training and following the first competitive cycle of the season. It is also our intent to describe structural tendon change, as well as to find associations between changes in tendon structure and jumping performance.

## 2. Materials and Methods

### 2.1. Settings and Participants

This was a cross-sectional study with a convenience sample of 19 volunteers of a male professional handball team with more than 7 years of handball practice, competing in national league and training for more than 25 h per week. All team players participating in training and matches during the first competitive cycle were included in the study. Participants were excluded if they were diagnosed with patellar tendinopathy in the last 12 months, had a medical history of patellar tendon rupture, or presented a time loss for injury or illness longer than 2 weeks. Approval of the study was obtained by the local ethical committee at Barça Innovation Hub (BiHub_HbB_2019-20), and all procedures followed the latest version the Declaration of Helsinki. Before commencing the study participants signed the Football club of Barcelona medical informed consent form for all the medical and fitness tests during the season. 

### 2.2. Variables and Measurements

For this study we included UTC scans in order to standardize regular training and performance testing. Before measures were taken, all participants underwent a 10 min standardized warm up, consisting of 8 min run, 4 × 12 squats, 3 × 10 lunge and 2 × 10 bilateral jump. UTC scanning was performed in each leg, by a single tester with more than five years of experience scanning and analysing with the UTC system and who performed all UTC scans in order to decrease possible inter-tester bias [18]. UTC scans were performed using B-mode ultrasound, with a linear transducer of 7–10 MHz (SmartProbe 10L5; Terason 2000, Teratech, Rockville, MD, USA). The ultrasound probe (SmartProbe 12L5-V, Terason 2000+; Teratech) was fixed to a tracking device (UTC Tracker, UTC Imaging) that automatically moved the transducer on the perpendicular axis of the tendon and recorded cross-images at 0.2 mm intervals [19]. The consecutive transversal images were then used to create 3D reconstructions. Intensity and distribution of grey images was calculated over 4.8 mm using UTC algorithms; window size 17 was used for imaging analysis. The region of interest (ROI) was located around the tendon in transverse view; contours of the ROI were drawn at the proximal tendon (20% tendon length), mid tendon (40% tendon length), and distal tendon (80% tendon length), as is previous practice in the literature [19]. Tendon length was determined from patellar inferior angle to the most proximal tibial tuberosity, while total tendon free length was also found. All tendon contours were marked by one investigator, and echo types were quantified through the UTC software itself (UTC 2010) [20].

A jumping performance test was carried out by means of counter movement jump (CMJ), using the Chronojump–Boscosystem 12-contact platform (Chronojumps, Spain). Players performed a bilateral vertical jump standing on the contact platform; following a signal, players had to squat and jump as high as possible with their hands at the waist, with the aim of reaching their maximal vertical height [3,21,22]. CMJ was performed three times, with a 30 s rest between repetitions, supervised by a strength and conditioning trainer with more than 10 years of testing CMJ with force platforms. Counter-movement mechanisms such as these require the action of lower limb muscle through the stretch–shortening cycle (use of the elastic phase [22]). 

### 2.3. Testing Procedure and Instrumentation

Bilateral UTC scans of patellar tendon and the jumping performance test (CMJ) were performed on three occasions: the first day of pre-season training, last day of pre-season training and following the first competitive cycle (Figure 1); measurements were taken at the training court.

### 2.4. Sample Size

A convenient sample size of 19 participants was chosen for the study. In previous research, UTC has examined deficient subjects in team sport populations using varying sample sizes, typically around 18 male participants [23]. In order to check whether this was an appropriate sample size for our study, we carried out a post hoc power analysis that will be reported further down. 

### 2.5. Signal Processing and Data Analysis

Four echo types can be discriminated based on consistency, with echo types I and II representing the most stable patterns and continual tendon structure, and echo types III and IV depicting less stable and more unstructured tendon fibers. For research purposes, the sum of echo type I and II percentages higher than 90% were considered a healthy tendon; consequently, percentages of echo type III and IV higher than 10% were regarded as an injured tendon [13], while poor-quality scans were excluded. 

Distribution of the sample was calculated for normality using the Shapiro–Wilk test, with the results revealing a non-parametric sample distribution. Descriptive data were presented as healthy tendon (% echo type I + echo type II) and injured tendon (% echo type III + echo type IV) for each tendon length measured (20%, 40%, 80%), as well as CMJ jump force for each limb. 

Data were initially examined for normality and descriptive statistics were calculated to identify changes in tendon structure and jumping performance between: (a) the first day of pre-season training versus the last day of pre-season training; and (b) the first day of pre-season training versus the end of the first competitive cycle. Wilcoxon signed-rank test and Spearman’s test were conducted to identify associations between tendon structure and CMJ performance at the first day of pre-season training, the last day of pre-season training and following the first competitive cycle. In addition, Hedges’ g was used to calculate the effect size, as the prior data sample differed from the latter one; significance at an alpha level of 0.05 was applied. All statistical calculations were performed with SPSS 26 (IBM, Amarok, USA) software.

## 3. Results

Nineteen male handball professional players started the study (Age = 20.3 ± 3.1 years; Height = 185.4 ± 6.18 cms; Weight = 84.32 ± 9.1 kg), of whom only 13 completed the data collection due to injury and time away from training. (Age = 20.1 ± 2.5 years; Height = 179.2 ± 3.36 cms; Weight = 81.32 ± 6.9 kg). Inter-rater reliability demonstrated moderate to excellent interclass correlation coefficient (ICC) of UTC [18]. Prior to the study, two independent raters performed UTC analysis of 20%, 40% and 80% of tendon length in ten patellar tendons, and a two-way mixed effect of absolute agreement was used to calculate the intraclass correlation coefficient (ICC) of each echotype [24]. The above Table 1 shows the intraclass correlation coefficient (ICC) values for patellar tendon UTC for all echotypes at 20%, 40% and 80% of tendon length. ICC for CMJ is also reported to be a reliable test; however, it was not performed in this study [25].

The descriptive data median, interquartile range (IQR), and coefficient interval percentage at 95% at the first and last day of pre-season and following the first competitive training for healthy and injured tendon at 20%, 40% and 80% tendon length is provided in the Table A1, Table A2 and Table A3, as well as in Figure 2. 

The results showed that all participants presented healthy tendon in the proximal, mid and distal tendons. 

The principal findings of the study showed structural changes in tendon, mainly at the healthy mid (*p* = 0.01) and distal tendon (*p* = 0.01) at the end of pre-season training. During pre-season, there were also structural alterations in injured tendon at the distal tendon (Right *p* = 0.02; Left *p* = 0.02) and proximal tendon (Right knee = 0.04) lengths.

At the end of the first competitive cycle, structural changes occurred only in injured tendon at the distal tendon length (*p* = 0.02). 

Descriptive data (Newtons) on jumping performance (CMJ), median and interquartile range, and coefficient interval at 95% at the first and last day of pre-season training and following the first competitive cycle are shown in Table 2.

Furthermore, the results showed no association between tendon structure and jumping performance at the first and last day of pre-season training, whereas an association appeared following the first competitive training between proximal tendon and jumping performance in the injured tendon (Right *p* = 0.04, Left *p* = 0.01) (see Table 3). 

The results of the study showed structural tendon changes occurring mainly at the end of pre-season training, highlighting a tendency to decrease in healthy tendon and to increase in injured tendon at the distal portion. Although the tendon structure was poorest, there were no associations between performance and tendon structure following pre-season training. 

Further changes were seen in the distal tendon following the first competitive cycle, despite the fact that the increase in injured tendon and jumping performance continued improving following the first competitive cycle. It is worth noting that this is an unexpected result, since there was an association between improved jumping performance and poor tendon structure at the end of the first competitive cycle. 

Post Hoc Sample Size Calculation

A post hoc power analysis estimated that, using a power of 80% and α = 0.05, 19 participants would have been needed to show a significant change in tendon structure for a healthy tendon (Hedge’s g = 0.97), while 22 participants would be adequate in the case of an injured tendon (Hedge’s g = 0.96).

## 4. Discussion

Patellar tendon structural changes are known to be adaptations to load over a given period. 

This study demonstrates tendon adaptations, mainly after pre-season training and following the first competitive cycles. Alterations in the tendon structure occur along all the tendon length, mostly at the mid and distal third of it. Nonetheless, tendon structure percentage remains within rates of healthy tendon, whereas the summatory of echo types I and II must be greater than 90% [14]. The results showed a tendency toward decrease in healthy tendon and increase in injured tendon (echo types III and IV), even though we were not able to demonstrate this tendency statistically. The results of this study support previous histological findings proving that tendon structural changes are a response to the load applied [25,26].

In our study, most changes occurred during pre-season (20% Inj_T Right, *p* = 0.01; 20 Inj_T Left, *p* = 0.04; 40% Inj_T Right, *p* = 0.02; 40% Inj_T Left, *p* = 0.01; 80% Inj _T Right, *p* = 0.02). This may indicate that fast increases in load demonstrate further negative tendon adaptation. Moreover, it could be suggested that during pre-season training, tendon physiological adaptations does not allow optimal homeostasis. When this balance is lost, apoptosis occurs as a result of poor tendon adaptation, impairing and the organized structure of the tendon [11]. It was also noted that during the first competitive cycle, a certain degree of adaptation persists. It is possible that once the season begins, in spite of regularity in the amount of training and competition loads, there may still be negative tendon adaptations. An increase in injured tendon (80% Inj_T Left, *p* = 0.02) similarly happened when load was consistent but remained at a very high intensity.

In volleyball, structural changes in the patellar tendon have also been observed during different times of the season [27]. Decreases in echo type I and increases in echo type II were observed in the dominant leg, while echo types III and IV showed no change [27]. In the present study, consistent with Rabello’s study (2019), echo types I and II were decreased in a follow-up at seven weeks of training [2].

In handball there is a continuous process of tendon adaptation to load, mainly when sudden or high intensity loads are applied; as demonstrated in this study, physiological structural changes in the tendon occur throughout the pre-season and first competitive cycle. These tendon adaptations over a period of time would seem to support the tendinopathy continuum concept [28], in which tendon structure changes occur even if pain is not present, remaining as asymptomatic disorganization of the tendon matrix [11,26,28]. Thus, the presence of disorganized tendon structure could take place prior to the onset of pain, while asymptomatic disorganized tendon structure may be a sign of potential tendinopathy.

In our study, there was a vague tendency toward increased injured tendon to the detriment of healthy tendon after a course of sudden and high intensity training. In spite of the belief in increased symptomatic tendinopathy associated with disorganized tendon structure, it is unclear how much tendon disorganization or percentage of tendon disorganization needs to occur in order for a tendon to become symptomatic. 

While our results, along with others, confirm that tendon response after high intensity activities shows changes in echo type and tendon structure [10,27,29], this is the only study reporting the location along the tendon where these changes occur. It is well known that patellar tendinopathy occurs mainly in the proximal third of the tendon, whether or not accompanied by tendon adaptation or pain [30,31]. In this study, structural tendon changes occurred at all three potions of the tendon, proximal, mid and distal; therefore, it could be suggested that tendon does not receive the same tension equally all along its length; consequently, changes in tendon structure could occur throughout the tendon.

This study also demonstrated structural changes in both legs regardless of the dominant side or jumping leg. Handball sport-specific mechanisms, such as unilateral jumping, acceleration, deceleration, change of direction, and sudden movements at the same position in the field, are asymmetrical. Although potential load imbalance due to sport demands could be presumed, this study was not able to demonstrate further structural tendon changes in the jumping or dominant leg. 

In addition, our study could not show associations between tendon structure and jump performance during pre-season. Associations were seen only in the injured tendon at 20% of length (Left *p* = 0.01; Right *p* = 0.04) following the first competitive cycle. Despite the structural tendon changes occurring over pre-season and the first competitive cycle, jumping performance continued to improve throughout the latter.

By the same token, while this study has been able to associate structural tendon changes in proximal injured tendon with increased jumping performance following the first competitive cycle, it should be noted that, while not statistically significant, there was a decrease in jumping improvement of the right limb, coincident to the leg with further structural disorganization. These results do not support the theory that organized tendon matrix enhances energy mechanical transduction, as jumping performance continued to improve while presenting slight tendon disorganization. The degree of tendon disorganization or percentage of disorganized tendon needed in order to be reflected in jumping performance is unknown; the possibility exists that jumping performance is hampered by pain rather than tendon structure, or that further tendon disorganization is needed before decreased jumping performance becomes apparent.

During handball season, high training loads and sport-specific demands produce adaptations in different anatomical structures [10]. Tendinopathy is diagnosed by structural change, which in turn presents as disorganized matrix, pain, and dysfunction. In this study, we demonstrated early negative tendon adaptation occurring during pre-season that could be related to sudden and high increased of training loads. According to these results, gradual onset of training load during pre-season training could allow the tendon to adapt, potentially preserving a higher percentage of healthy tendon. Although structural changes were shown, there was no association with jumping performance, and further studies are required to quantify the amount of disorganized tendon needed to decrease performance or to become symptomatic. Quantifying the quality of tendon structure allows treatment and prevention of the of future tendinopathy and detection of when a decrease in mechanotransducer properties is enough to decrease performance.

Several limitations have been detected in this study. The small sample size of the study increases the chance of obtaining a Type II error. Symptomatic player registration would have been important information related to changes in tendon structure. In future studies it is recommended to use more parameters to define the performance variables, as well as to monitor GPS training loads and matches, quadriceps strength, and pain level, if present. Taking these limitations into consideration, further studies are needed in order to understand the mechanisms related to tendon structure change and its association with load, individual response and performance.

## 5. Conclusions

Patellar tendon structural changes were observed mainly during pre-season training throughout the first competitive cycle in male professional handball players. Changes occurred in all portions of the patellar tendon, although proximal and distal tendon showed greater change than mid tendon throughout the pre-season and first competitive cycle. Although changes in tendon structure were seen at the end of the first competitive cycle, a mild decrease in healthy tendon and increase in injured tendon was seen. In addition, an association was found between tendon structural changes and jumping performance in the proximal tendon of the dominant leg. We were not able to find an association between jumping performance and changes in tendon structure. 

## Figures and Tables

**Figure 1 ijerph-18-12156-f001:**
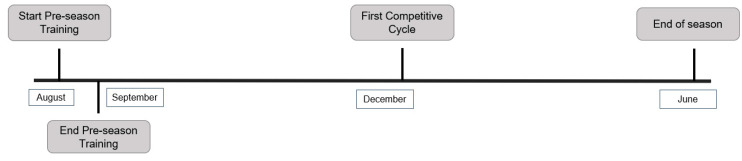
Data Collection Diagram.

**Figure 2 ijerph-18-12156-f002:**
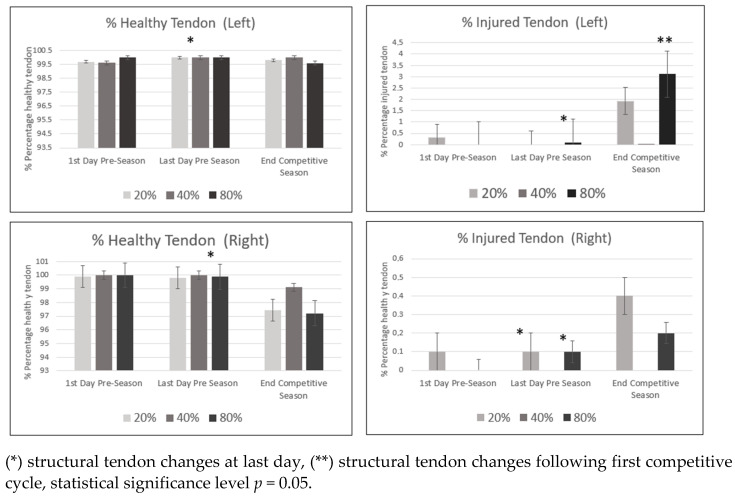
Percentage of healthy and injured tendon (median and Inter quartile Range (IQR) at 20%, 40% and 80% tendon length) at the first and last day of pre-season and following the first competitive training.

**Table 1 ijerph-18-12156-t001:** Interclass correlation coefficient of UTC scan of the patellar tendon.

*Echotype*	ICC (95% CI)		
	20%	40%	80%
** *I* **	0.93 (0.74–0.98)	0.97 (0.91–0.99)	0.88 (0.54–0.97)
** *II* **	0.84 (0.42–0.96)	0.90 (0.65–0.98)	0.80 (0.23–0.95)
** *III* **	0.81 (0.24–0.95)	0.84 (0.38–0.96)	0.89 (0.60–0.97)
** *IV* **	0.88 (0.55–0.97)	0.90 (0.63–0.98)	0.82 (0.33–0.95)

**Table 2 ijerph-18-12156-t002:** Jumping performance descriptive data (median and interquartile range, coefficient interval at 95%) at the first and last day of pre-season training and following first competitive cycle.

CMJ (Newton)		First Day Pre-Season Training		Last Day Pre-Season Training		Following First Competitive Cycle	
		Left	Right	Left	Right	Left	Right
Median (IQR)		757.5 (801.5)	748.3 (819.3)	794.5 (867.2)	812.2 (847.9)	848.7 (913.8)	855.9 (896.5)
CI (95%)	Inf Lim	731.8	744.1	735.6	750.3	784.4	790.4
	Sup Lim	825.0	820.9	848.5	851.1	920.3	904.8

Descriptive data (Newtons). of counter movement jump (CMJ), (median, inter quartile Range (IQR). coefficient Interval (CI) at 95%. Inferior Limit (Inf Lim). Superior Limit (Sup Lim).

**Table 3 ijerph-18-12156-t003:** Association between tendon structure and jumping performance (CMJ) (Hedges’ g effect size and *p*-value) at the first and last day of pre-season training and following the first competitive cycle.

Association between Tendon Structure and CMJ (Mean Right and Left)						
Distance/Tendon Type	First Day Pre-Season Training		Last Day Pre-Season Training		Following First Competitive Cycle	
	Effect Size g	*p*-Value	Effect Size g	*p*-Value	Effect Size g	*p*-Value
20_Healthy_T Left (%)	−0.98	0.73	−0.97	0.86	−0.98	0.94
20_Healthy_T Right (%)	−0.97	0.77	−0.98	0.96	−0.98	0.96
40_Healthy_T Left (%)	−0.98	0.53	−0.98	0.43	−0.98	0.56
40_Healthy_T Right (%)	−0.97	0.48	−0.98	0.77	−0.98	0.71
80_Healthy_T Left (%)	−0.98	0.67	−0.97	0.94	−0.98	0.85
80_Healthy_T Right (%)	−0.97	0.98	−0.98	0.33	−0.98	0.84
20_Injured_T Left (%)	−0.97	0.64	−0.98	0.70	−0.98	0.01
20_Injured_T Right (%)	−0.98	0.54	−0.98	0.68	−0.98	0.04
40_Injured_T Left (%)	−0.98	0.98	−0.98	0.75	−0.98	0.21
40_Injured_T Right (%)	−0.98	0.52	−0.98	0.40	−0.98	0.45
80_Injured_T Left (%)	−0.97	0.75	−0.98	0.98	−0.98	0.98
80_Injured_T Right (%)	−0.98	0.98	−0.98	0.45	−0.98	0.79

CMJ (Counter movement jump), Tendon (T), (%) Percentage.

## Data Availability

The data presented in this study are available on request from the corresponding author.

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
