# Peer review of "Patellar Tendon Structural Adaptations Occur during Pre-Season and First Competitive Cycle in Male Professional Handball Players"

_ijerph, 2021, doi:10.3390/ijerph182212156_

Round 1

Reviewer 1 Report

This study investigates changes in patellar tendon structure during pre-season training and at the end of the first competitive cycle, and associations between tendon structure and jumping performance. The authors found changes in patellar tendon structrure after pre-season training. However, there were no associations between changes of tendon structure and CMJ performance.

The scope and findings of the study are in my opinion of relevance to sports science and public health, however it is not related to the environmental research.

There are minor concerns which should be addressed.

Please, include short conclusion and eventually applications of findings in the practice in the abstract.

Page 1, lines 30-32: Patellar tendon structrure shows changes after pre-season training. No associations were found between changes of tendon structure and CMJ performance.

Consider to shortening the introduction.

Page 1, lines 14-18: High training loads and sport specific demands predispose handball players from suffering patellar tendinopathy, mainly during the pre-season training which continued throughout the season. Ultrasound Tissue Characterization (UTC) can quantify tendon structure adaptations and changes. It is uncertain how patellar tendon structure changes to training loads during season and how it affects to performance in handball professional player.

Consider to reformulate the second aim of the study and set up new hypothesis (or formulate research questions).

Pages 2-3, lines 96-99: This study aims to identify changes in tendon structure after pre-season training and following the first competitive cycle of the season. In addition, the study intends to find possible associations between tendon structure and jumping performance during the same period of time.

Specify inclusion criteria for handball players to be allocated to the study (their performance level, years of training, etc.).

Page 3, lines 103-107: For the study, all players of the team who participated in trainings and matches during the first competitive cycle were included. Participants were excluded if they were diagnosed with patellar tendinopathy in the last 12 months, medical history of patellar tendon rupture, or presented a time loss for injury or illness longer than 2 weeks.

Include description of axes in six figures on page 9.

The discussion reflects what authors found and how it relates to the literature. However, they should present the practical applications of obtained findings with respect to a specific group of handball players.

Author Response

Thank you for the reviewer’s comments

The reviewer is right, the study relates to sport science and public health, this why we thought the article could be published in Exercise and Health section  in the IJERPH. From our understanding Exercise Health will have a special issue;”Neuromuscular Performance and Wellness in Sport: Assessment, Training, Gender and Technology”. Our article relates to sport, assessment training, gender and technology, we believe our paper could be very suitable for this journal. We also believe that evaluating damage or potential damage to the “environment” in our case humans (handball players) that need to be protected we are relating to environmental research. We appreciate the reviewer comments as it has been really useful to know more about the meaning of environmental research and a new project has come up from the reviewer comment.

Reviewer 2 Report

The manuscript “Patellar tendon structural adaptations occur during pre-season and first competitive cycle in male professional handball players” authored by Ortega-Cebrian et al. represents an interesting work to evaluate the changes occurred at the patellar tendons during pre-season training and at the end of the first competition. The study sounds interesting, however, the authors are invited to answer and adjust the quality of the manuscript before being considered for publication.

  1. Abstract

The abstract is poorly written, and the authors should render it more concrete and attractive for the readers.

  1. Introduction
  • In the sentence “Epidemiological studies report time loss of sport practice, but not presence symptomatic pathology in active athletes, which frequently occurs in player diagnosed with patellar tendinopathy.”, the sentence is not clear please rephrase it.
  • The whole introduction does not follow a chronological order and the information are somehow confusing. Please readjust the introduction part.
  1. Materials and Methods.
  • Figure 1: Adjust “Juny” to “June”.
  1. Results
  • What does “ICC” mean? Please explain.
  • The authors are invited to adjust the Table 3 representing the changes occurred after the last day pre-season and 1st competitive cycle. The results should be explained clearly and well detailed within the manuscript. The authors should try to render the presentation of the results easier to read. Maybe it would be better to put all data also concerning the 1st day pre-season in terms of effect size and p-value together to better understand the obtained results.
  • The results within the Tables should be better explained since it is confusable to see “effect size”, “p-value” in every single Table. The authors are invited to reorganize the table to facilitate data interpretation. I think the Table should be transformed into graphs Like Figure 2 in order to render them easy to understand and more reasonable.
  1. Discussion

The overall discussion should be rephrased and well structured.

Author Response

The article has been peer reviewed for English grammar, as well as written style have been corrected. Please, fins attached further corrections 

Reviewer 3 Report

The work is interesting. It contains numerous editorial errors. I suggest the authors read https://www.mdpi.com/journal/ijerph/instructions. A few things need clarification (see Substantive errors).

Substantive errors

L5 - Different author names in the mpdi system and manuscript.

mpdi system - Silvia Ortega-Cebrián , Ramon Navarro , Sergi Seda , Sebastia Salas , Myriam Guerra-Balic

manuscript - Ortega-Cebrian, Silvia, Ramon Navarro Gonzalez  , Sergi Seda ,Sebastià Salas, Miriam Guerra

L34 - " ultrasound tissue characterization", " load"," tendon structure" - Keywords are not MeSH compliant. Change according to MeSH. https://meshb.nlm.nih.gov/search.

L37 - " Patellar tendinopathy" - add statistical data on this injury. On the scale of the sport in question and on a global scale.

L103 - " professional" - this can be understood in different ways, describe exactly what you mean by professional.

L107-108 - add the number of the approval of the study protocol by the bioethics committee.

L113-L114 - "Before  measures  were  taken,  all  participants  underwent  10  min  standardized warm up." - describe what you mean by a standardized warm-up.

L400-401 - "Percentages of healthy tendon remains within healthy parameters (echo type I and II above 90%)" - for me it's repeat results - reformulate or delete.

L418-422 - add Bioethics Committee approval number.

Editorial errors

L 5 "Ortega-Cebrian, Silvia" - remove the comma;

L5 "Sergi Seda2 ,Sebastià Salas2" - add spaces after the comma;

L7 - "C/" - what does that mean?

L8 -"F.C." - expand the shortcut.

L9 - ", s/n. AV." - what does that mean?

L9 -  "Sant Joan Despí;;" - remove one semicolon.

L17 - "changes.    It is uncertain" - remove double space.

L42 - change "teams[1]." to "teams [1]" - space and quotation - the comment applies throughout the text. Add spaces between words and quotes.

L70 - "MRI" - expand the abbreviation when you first use it.

L104 - "  included.    Participants  were" - - remove double space.

L109 - "FCB Medical" - Expand the shortcut.

L112 - "UTC" Expand the shortcut. Expand the abbreviation in the main text as you use it for the first time. You separately expand a shortcut for the abstract and for the main text.

L135 -"[use of the elastic phase)[20].  " - Unify parentheses.

L137-" test  [CMJ)  were" - Unify parentheses.

Table 2. - the first column deactivates bold. This is not consistent with mpdi tempel (see https://www.mdpi.com/journal/ijerph/instructions)

L210 - change "Table 3:" to "Table 3."

L210-211 - "Changes in tendon structure since 1st day pre-season training to last day pre-season trainign  and following fisrt competitive cycle" - Why did you bold the text? deactivate bold. This is not consistent with mpdi tempel

Table 3 - the first column deactivate bold. This is not consistent with mpdi tempel.

L238-239 - A different font is used for the caption of Table 2 and another here. Unify. In accordance with IJERPH standards, suggests font Palatino Linotype size 9

L245 - change "( p= 0.02)" to "(p= 0.02)"

Table 4 - the first column deactivates bold. This is not consistent with mpdi tempel (see https://www.mdpi.com/journal/ijerph/instructions)

L290-291- A different font is used for the caption of Table 2 and another here. Unify. In accordance with IJERPH standards, suggests font Palatino Linotype size 9

L297 - change  "Table 5:" to "Table 5."

Table 5 -" Table 5: Descriptive and Analytic Data of Healthy, Injured Tendon Structure and CMJ Perfor-mance Following 1srt Competitive Cycle. "- Why did you bold the text? deactivate bold. This is not consistent with mpdi tempel.

L303-304- A different font is used for the caption of Table 2 and another here. Unify. In accordance with IJERPH standards, suggests font Palatino Linotype size 9

L317 - delete "Inset"

L317 - change "Figure 2:" to "Figure2."

L317-318 - According to the mpdi temple, the caption should be under the graphic. (see https://www.mdpi.com/journal/ijerph/instructions - Submission Checklist - "use the Microsoft Word template or LaTeX template or Free Format Submission to prepare your manuscript;")

L322- Unify. In accordance with IJERPH standards, suggests font Palatino Linotype size 9.

L325 - " using a power of 80%  and" - remove double space.

L327 - change " ( Hedge’s g = 0.97 )" to "(Hedge’s g = 0.97)"

L406-408 - delete " For research articles with several authors, a short paragraph specifying their individual contributions must be provided.  The  following  statements should be used"

L408-413 - remove the quotation marks

L408-413 / L10 - initials SS and SS. It is not clear which author is meant. For example, add the next letter of your last name (Sergi Seda - SSe and Sebastià Salas - SSa).

L419 - Change "FCBarcelona" to "FC Barcelona" or use the full name

L420 - Change "FCBarcelona" to "FC Barcelona" or use the full name

L427 - add period.

L429- add period.

L430 - 497 - no item in the bibliography is written to the journal's standard.

"References should be described as follows, depending on the type of work:

ï‚·  Journal Articles:
1. Author 1, A.B.; Author 2, C.D. Title of the article. Abbreviated Journal Name Year, Volume, page range.

(...)

Websites:

  1. Title of Site. Available online: URL (accessed on Day Month Year).

Unlike published works, websites may change over time or disappear, so we encourage you to create an archive of the cited website using a service such as WebCite. Archived websites should be cited using the link provided as follows:

(...)" - (see https://www.mdpi.com/journal/ijerph/instructions)

The entire bibliography is up for improvement.

I mark the major mistakes:

L438 -  lack of pages.

L460 - lack of pages.

L468 - lack of pages.

L430 - 497 - in addition to not using the mpdi citation style, the authors mix different citation styles. Such a reference is unacceptable.

Thank you for the opportunity to review this work.

Author Response

We appreciate the detailed correction, all substative, editorial errors have been corrected. Reference follow ACS reference style. Corrections in the manuscript are mentioned in the below answers

Round 2

Reviewer 3 Report

The version I received is unreadable because of the amount of editing. I would suggest in the future just marking the text added with a different color.

Congrats to the authors for the work they put into improving the manuscript.

Substantive errors - have been corrected according to my suggestions.

Editorial errors - Despite the authors' assurances, some comments have not been corrected and new editorial errors have appeared.

My sample comments that have not been corrected:

L406-408 - delete " For research articles with several authors, a short paragraph specifying their
individual contributions must be provided. The following statements should be used"

 L408-413 / L10 - initials SS and SS. It is not clear which author is meant. For example, add the next  letter of your last name (Sergi Seda - SSe and Sebastià Salas - SSa).

L427 - add period.
We were not able to identify this comment

L689- change "this paper" to "this paper."

L429- add period.
We were not able to identify this comment

L691 - change "The authors declare no conflict of interest" to "The authors declare no conflict of interest."

References is still not correct, i don't understand why the authors changed the citations to superscript.

"In the text, reference numbers should be placed in square brackets [ ], and placed before the punctuation; for example [1], [1–3] or [1,3]. For embedded citations in the text with pagination, use both parentheses and brackets to indicate the reference number and page numbers; for example [5] (p. 10). or [6] (pp. 101–105)."

The authors have not read Instructions for Authors -https://www.mdpi.com/journal/ijerph/instructions as I suggested.

L560 - change "p=0.04;" to "p= 0.04"

In conclusion, Substantive errors have been corrected, editorial errors have not been corrected, new ones have appeared. Due to the current form of the work - lack of readability, I am not able to mark the remaining editorial errors. Authors don't know https://www. mdpi. com/journal/ijerph/instructions

Thank you for the opportunity to review this work.

Author Response

Reviewer 3

Review 2 (minor revision)

Comments and Suggestions for Authors

 Thank the reviewr for considering the paper for minor revisions

The version I received is unreadable because of the amount of editing. I would suggest in the future just marking the text added with a different color.

Congrats to the authors for the work they put into improving the manuscript.

We thank the reviewer for the comments and suggestions to use different colors to added text.

Substantive errors - have been corrected according to my suggestions.

Editorial errors - Despite the authors' assurances, some comments have not been corrected and new editorial errors have appeared.

The reviewer is right, authors have mixed different manuscripts, we have corrected abstract, further editorial error according to the journal instructions

My sample comments that have not been corrected:

L406-408 - delete " For research articles with several authors, a short paragraph specifying their
individual contributions must be provided. The following statements should be used"

The sentence finally deleted

 L408-413 / L10 - initials SS and SS. It is not clear which author is meant. For example, add the next  letter of your last name (Sergi Seda - SSe and Sebastià Salas - SSa).

WE have added a changes as Sergi Seda-López ( SSL) marked with “Track Changes”. Page 1 lines 19, page 3 line 69 and page 14 line 548.

L427 - add period.
We were not able to identify this comment

Text added: throughout pre-season and first competitive cycle. page 11 line 445

L689- change "this paper" to "this paper."

Done and mark with “Track Changes

L429- add period.
We were not able to identify this comment

Text added: “at the end the first competitive cycle” page 11 line 446

L691 - change "The authors declare no conflict of interest" to "The authors declare no conflict of interest."

Done and mark with “Track Changes

References is still not correct, i don't understand why the authors changed the citations to superscript.

"In the text, reference numbers should be placed in square brackets [ ], and placed before the punctuation; for example [1], [1–3] or [1,3]. For embedded citations in the text with pagination, use both parentheses and brackets to indicate the reference number and page numbers; for example [5] (p. 10). or [6] (pp. 101–105)."

 We have reviewed the reference and added the []

The authors have not read Instructions for Authors -https://www.mdpi.com/journal/ijerph/instructions as I suggested.

L560 - change "p=0.04;" to "p= 0.04"

We have found the same error in line 404 page 10

 In conclusion, Substantive errors have been corrected, editorial errors have not been corrected, new ones have appeared. Due to the current form of the work - lack of readability, I am not able to mark the remaining editorial errors. Authors don't know https://www. mdpi. com/journal/ijerph/instructions

We have reviewed https://www.mdpi.com/journal/ijerph/instructions thoroughly, we have changes in the abstract as well as further editorial error. We hope the corrections present the article ready

Thank you for the opportunity to review this work.
